# Antimicrobial Activity of Essential Oils Evaluated In Vitro against *Escherichia coli* and *Staphylococcus aureus*

**DOI:** 10.3390/antibiotics11070979

**Published:** 2022-07-20

**Authors:** Michela Galgano, Paolo Capozza, Francesco Pellegrini, Marco Cordisco, Alessio Sposato, Sabina Sblano, Michele Camero, Gianvito Lanave, Giuseppe Fracchiolla, Marialaura Corrente, Francesco Cirone, Adriana Trotta, Maria Tempesta, Domenico Buonavoglia, Annamaria Pratelli

**Affiliations:** 1Department of Veterinary Medicine, University Aldo Moro of Bari, 70010 Valenzano, Italy; michela.galgano@uniba.it (M.G.); paolo.capozza@uniba.it (P.C.); francesco.pellegrini@uniba.it (F.P.); marco.cordisco@uniba.it (M.C.); alessio.sposato@izspb.it (A.S.); michele.camero@uniba.it (M.C.); gianvito.lanave@uniba.it (G.L.); marialaura.corrente@uniba.it (M.C.); francesco.cirone@uniba.it (F.C.); adriana.trotta@uniba.it (A.T.); maria.tempesta@uniba.it (M.T.); domenico.buonavoglia@uniba.it (D.B.); 2Department of Pharmacy-Drug Sciences, University Aldo Moro of Bari, 70125 Bari, Italy; sabina.sblano@uniba.it (S.S.); giuseppe.fracchiolla@uniba.it (G.F.)

**Keywords:** *Escherichia coli*, *Staphylococcus aureus*, essential oils, GC-MS analysis, antimicrobial activity

## Abstract

The spread of extended-spectrum β-lactamase-producing *Escherichia coli* and methicillin-resistant *Staphylococcus aureus* has caused a reduction in antibiotic effectiveness and an increase in mortality rates. Essential oils (EOs), known for their therapeutic efficacy, can be configured as novel broad-spectrum biocides. Accordingly, the bacteriostatic–bactericidal activity of *Citrus Lemon* (LEO), *Pinus Sylvestris* (PEO), *Foeniculum Vulgaris* (FEO), *Ocimum Basilicum* (BEO), *Melissa Officinalis* (MEO), *Thymus Vulgaris* (TEO), and *Zingiber Officinalis Rosc.* (GEO), at concentrations ranging from 1.25 to 40% (*v*/*v*), were tested in vitro against different *E. coli* and *S. aureus* strains using minimal inhibitory concentrations (MICs) and minimum bactericidal concentrations (MBCs). The chemical compositions of the EOs were analyzed using GC/MS. The major components of all seven tested oils were limonene, α-pinene, anethole, estragole, citral, thymol, and zingiberene, respectively. We found that the bacteriostatic–bactericidal activity of the EOs was related to their chemotypes and concentrations, as well as the strain of the bacteria. A dose–effect correlation was found when testing GEO against *S. aureus* strains*,* whilst FEO was found to have no activity regardless of concentration. PEO, MEO, and BEO were found to have bactericidal effect with a MIC and MBC of 1.25% (*v*/*v*) against *S. aureus* strains, and LEO was found to have values of 1.25% (*v*/*v*) and 5% (*v*/*v*) against ATCC and clinical isolate, respectively. Interestingly, the antimicrobial activity of TEO was not related to oil concentration and the complete inhibition of growth across all *E. coli* and *S. aureus* was observed. Although preliminary, our data demonstrate the efficacy of EOs and pave the way for further investigations on their potential synergistic use with traditional drugs in the human and veterinary fields.

## 1. Introduction

The indiscriminate use of antibacterial agents for therapeutic and prophylactic purposes in a variety of fields, e.g., animal husbandry and agriculture, has led to antibiotic resistance to a degree now regarded as one of the greatest global health problems [1]. Several bacteria genera, with the most common being *Escherichia coli* and *Staphylococcus aureus*, have developed multidrug resistance [2].

*E. coli*, Gram-negative *Enterobacteriaceae* found mainly in the gastrointestinal tract of various species of domestic and wild animals and in the environment, e.g., in soil, water, and plants, can cause mild to severe infections, leading to death by septicemia. *E. coli* is responsible for gastrointestinal as well as urinary tract infections and other local tissue and organ infections [3]. *Staphylococcus* spp. are commensal bacteria living on the skin and mucous membranes of various hosts and often act as opportunistic pathogens. These bacteria are considered the second-most-common pathogen reported among elderly patients hospitalized with bacteremia at a recuperation or convalescent facility. In particular, *Staphylococcus aureus* is the fourth-most-common hospital-acquired pathogen among older adults, following *E. coli*, *Pseudomonas aeruginosa,* and enterococci and accounting for 9% of all nosocomial infections [4]. In recent years, veterinary medicine has also seen increased rates of antimicrobial resistance genes in commensal bacteria, i.e., *E. coli* and *Staphylococcus* spp., which have been repeatedly observed in dog and cat populations. This phenomenon has been associated with possible treatment failures, longer periods of hospitalization, increased costs for treatments, and morbidity [5]. *E. coli* is a representative indicator of antimicrobial resistance (AMR) in Gram-negative bacteria [6], whilst *Staphylococcus* spp., in particular methicillin-resistant *S. aureus* (MRSA), is the focus of the AMR surveillance program in food-producing animals [7]. Therefore, the spread of these multidrug-resistant (MDR) bacteria constitutes a serious public health threat due to potential interspecies transmission, including with humans [8].

In this context, it is extremely important to identify new natural molecules that can act as antibacterial agents as well as guaranteed safety and non-toxicity. Essential oils (EOs), also known as volatile oils, are products of the secondary metabolism of aromatic plants. EOs display different biological properties, such as anti-inflammatory, sedative, digestive, antimicrobial, antiviral, or antioxidant activities [9,10], and have been widely employed as excellent substitutes for chemical-based preservatives. Previous studies confirmed the antibacterial action of EOs against *S. aureus* [11], *Enterobacter aerogenes*, *Klebsiella oxytoca*, *Proteus mirabilis*, *Morganella morganii* and *E.*
*coli* [12]. Moreover, bactericidal activity and the ability of EOs to interfere with bacterial replication have prompted their use in the pharmaceutical, health, cosmetic, agricultural, and food industries [13].

The antimicrobial activity of EOs is highly variable, mainly influenced by their chemical composition and/or concentration [14]. Some EO components act on the lipid bilayer of the cell membrane, others negatively affect the cell cycle (S phase) of bacteria, and others inhibit protein synthesis and DNA replication [15]. There are several hypotheses on the antimicrobial (bactericidal and/or bacteriostatic) properties of the different parts of an EO, and whether individual components (e.g., limonene, geranial, p-cymene) are more effective than whole oils (e.g., lemon oil, lemon balm oil, thyme oil) is currently not clear [16].

This study aims to define the chemical composition of different commercial EOs and to evaluate the potential antibacterial activity of different EOs against two strains of *S. aureus* (ATCC 11622 and urinary bacterial isolate) and two strains of *E. coli* (ATCC 25922 and urinary bacterial isolate). 

## 2. Results

### 2.1. Chemical Composition of the EOs

A complete description of all EO components is reported in Appendix A. Brief descriptions of the main components of all the EOs that we analyzed are reported in Table 1.

The analysis of *Citrus Lemon* (LEO) EO revealed a complex mixture consisting mainly of oxygenated and hydrocarbon monoterpenes. The six major detected compounds were limonene (53%), β-pinene (14.5%), γ-terpinene (5.9%), citral (3.8%), α-pinene (2.4%), and β-thujene (1.94%).

The dominating *Pinus Sylvestris* (PEO) EO compounds we detected were α-pinene (29%), β-pinene (17.2%), 3-carene (13.1%), limonene (9.8%), and bornyl acetate (5.7%). The main sesquiterpenes detected were caryophyllene (4.9%) and caryophylleneoxyde (1.68%).

The EO of *Foeniculum Vulgaris* (FEO) was composed of a mixture of several monoterpenes and phenylpropanoids, with anethole (58.7%), α-pinene (6.4%), α-phellandrene (5.2%), and limonene (5.1%) as the main constituents.

The leading compounds identified in the *Ocimum Basilicum* (BEO) EO were monoterpenes comprised of oxygenated and hydrocarbon monoterpenes, followed by phenylpropanoids and sesquiterpenes, such as oxygenated and hydrocarbon sesquiterpenes; this characterized about 98% of the whole mixture. The main constituents recognized were estragol (73%) and β-linalool (17%). Other identified constituents were α-bergamotene (3.2%) and citral (about 1.0 %).

About thirty compounds were identified in the EO of *Melissa officinalis* (MEO), accounting for 91.64% of the total. We found that this oil was characterized by the presence of citral (43%), caryophyllene (25%), humulene (4.4%), limonene (4.3%), cariophyllene oxide (2.2%), geranyl acetate (1.95%), and eucalyptol (1.2%).

The chemical composition of the *Thymus Vulgaris* (TEO) EO was determined using GC/MS. About twenty-five components were identified and composed 98.7% of the total detected constituents, as reported in a previous work [19]. The major components were thymol (47%), o-cymene (19.6%), and γ-terpinene (9%), which suggests that the EO analyzed belongs to the thymol chemotype. 

A total of 105 components were identified in the *Zingiber Officinalis Rosc* (GEO) EO, with 38 components corresponding to 92.85% of the mixture. Zingiberene was the predominant component, accounting for 32% of the mixture as reported in our previous work [10].

### 2.2. Antibacterial Activity

In the disc diffusion test, *E. coli* isolate #462/20 was found to be resistant to oxacillin, ampicillin, and erythromycin and *S. aureus* isolate #463/20 were found to be resistant to oxacillin and ampicillin.

The bactericidal and/or bacteriostatic properties of the seven tested EOs were evaluated by determining the minimal inhibitory concentration (MIC) and minimum bactericidal concentration (MBC). Antibacterial activity was correlated with the type and concentration of EO and with the bacterial species and strain. The correlation between the concentrations of the EOs and the number of colony-forming units (CFUs) /mL of bacteria tested was determined by calculating the Spearman coefficient (rho); a negative monotone relationship was identified, as evidenced by the scatter plot. The analysis shows that a reduction in the concentration of EOs was correlated with an increase in CFU/mL of the distinct bacterial strains under study. 

All tested EOs but TEO demonstrated low antimicrobial activity against both *E. coli* organisms. Therefore, the MIC and MBC could not be assessed. An inverse statistically significant correlation was observed between the different concentrations of BEO (rho: −0.98, *p*-value: 0.0003; rho: −0.94, *p*-value: 0.0016), GEO (rho: −0.98, *p*-value: 0.0003; rho: −0.92, *p*-value: 0.0076), FEO (rho: −1, *p*-value: 0.0027; rho: −1, *p*-value: 0.0027), LEO (rho: −1, *p*-value: 0.0027; rho: −1, *p*-value: 0.0027), MEO (rho: −0.94, *p*-value: 0.0050; rho: −1, *p*-value: 0.0027), and PEO (rho: −1, *p*-value: 0.0027; rho: −0.94, *p*-value: 0.0016) and the numbers of CFU/mL of both strains of *E. coli* (ATCC and clinical isolate #462/20, respectively). This correlation was detected down to the lowest concentration, 1.25% (*v*/*v*), at which a reduction in bacterial growth of at least 10^3^ CFU/mL for *E. coli* ATCC (when compared with the starting concentration of 1 × 10^16^ CFU/mL) and of 10^1^ CFU/mL for *E. coli* clinical isolate #462/20 (when compared with the starting concentration pf 1 × 10^14^ CFU/mL) was observed. The antimicrobial activity of TEO was not related to its concentration, resulting in a complete inhibition of the growth of both strains of *E. coli* even at the lowest concentrations (Table 2, panels a, b; Figure 1a,b). 

Low concentrations of PEO, BEO, MEO, and TEO demonstrated bactericidal effect, with a MIC and MBC of 1.25% (*v*/*v*) against both *S. aureus* strains, and the antimicrobial activity of these EOs was not correlated with different EO concentrations (*p*-value > 0.05). Bacterial growth was also strongly inhibited by LEO, with a MIC and MBC of 5% (*v*/*v*) for clinical isolate #463/20 and a MIC and MBC of 1.25% (*v*/*v*) for the ATCC strain. On the contrary, FEO did not demonstrate any antimicrobial activity, and MIC and MBC could not be evaluated. Regarding the correlation between the concentrations of the EOs and the number of CFU/mL, we found a statistically significant inverse correlation between all the tested concentrations of GEO (rho: −0.98; *p*-value: 0.0003) and the numbers of CFU/mL of the *S. aureus* ATCC strain. Moreover, a similar correlation was found for GEO (rho: −0.94; *p*-value: 0.005) and LEO (rho: −0.85; *p*-value: 0.03) with clinical isolates #463/20 of *S. aureus*. In particular, the antimicrobial activity of GEO against both *S. aureus* strains tested seemed to be greatly influenced by EO concentration, and a reduction of 10^1^ CFU/mL (from 1 × 10^13^ to 5 × 10^12^ CFU/mL) was observed when passing from a concentration of 5% (*v*/*v*) to 10% (*v*/*v*), and a further decrease of 10^1^ CFU/mL (from 5 × 10^14^ to 3 × 10^13^ CFU/mL) when EO concentration was halved from 2.5% (*v*/*v*) to 5% (*v*/*v*). No bacterial growth was observed when testing LEO with clinical isolate #463/20 at concentrations ranging from 40% to 5% (*v*/*v*), but concentrations of 2.5% (*v*/*v*) and 1.25% (*v*/*v*) allowed a growth of 10^12^ CFU/mL and 10^13^ CFU/mL, respectively. On the contrary, when LEO was tested with *S. aureus* ATCC, no bacterial growth was observed regardless of the concentration of LEO (Table 2, panels c, d; Figure 1c,d).

## 3. Discussion

The chemical compositions and antibacterial activity of seven commercially available EOs were tested. As shown in Table 1, most of the analyzed EOs contained some common compounds such as α-pinene, β-pinene, camphene, limonene, caryophyllene, eucalyptol, α -bergamotene, or caryophyllene oxide. Some of these compounds were present in low concentration in some EOs while being characteristic of others because of their large amount. Furthermore, other compounds were typical of a single EO; in particular, α-curcumene, zingiberene, and β-sesquiphellandrene were present only in GEO, anethole was reported for FEO only, and 3- carene for PEO.

Overall, a dose-dependent effect on bacterial growth in Gram-negative bacteria was observed for all the tested EOs and a bactericidal effect in Gram-positive bacteria was observed for most of them. This difference is explainable with the greater resistance of Gram-negative bacteria, given their outer membrane that acts as a barrier as well as the high content of cyclopropane fatty acids (CPA) in their cytoplasmic membranes [20]. On the other hand, in our study, TEO demonstrated overall bactericidal activity with complete growth inhibition of both Gram-positive and Gram-negative bacteria, regardless of the compound’s concentration [21,22,23,24]. Though the different chemical compositions of the various EOs and the lack of a standard for the evaluation of antimicrobial activity make it difficult to compare the several results previously reported [21,22,24], the antimicrobial activity of TEO is well known in the literature, as it can inhibit the growth of different bacterial species, both Gram-positive and Gram-negative [21,22,24].Interestingly, this effect does not appear to derive from its main component o-cymene (from 8.41% to 53.85%), which instead improves the antimicrobial properties of other substances, such as the monoterpenes α-pinene and β-pinene [25].

All the other EOs demonstrated a dose-dependent reduction in the growth of both strains of *E. coli* (ATCC and clinical isolate #462/20). In addition to TEO, PEO, BEO, and MEO also demonstrated bactericidal activity against both *S. aureus* strains, with non-dose-dependent growth inhibition. Among all the tested EOs, only GEO reduced the bacterial growth of all the tested strains in a dose-dependent manner. Its antimicrobial activity can be attributed to two components, monoterpenes and sesquiterpenes, as they alter the permeability and the fluidity of the plasma membrane of microorganisms [26]. On the other hand, FEO demonstrated no antibacterial activity towards either. 

LEO demonstrated a total inhibition of *S. aureus* ATCC growth, and a dose-related growth reduction of the clinical isolate #463/20 was observed when passing from a concentration of 5% (*v*/*v*), with total inhibition, to 2.5% (*v*/*v*) and 1.25% (*v*/*v*), with dose-related growth reduction. This is in accordance with the increase in bacterial permeability caused by LEO, which can cause the death of the bacteria. The reduction of the bactericidal effect at the above (sublethal) concentrations could be due to the response of microorganisms to the insult of EO on the cell membrane: reacting by increasing the expression of stress response proteins so as to repair the damaged proteins [27].

Interestingly, FEO exhibited antibacterial activity closely related to concentration when tested against both strains of *E. coli*, while no antibacterial activity was observed against both *S.*
*aureus* strains, in disagreement with data present in other studies [28]. The anethole-dominated FEO is considered as one of the two reference chemotypes accepted by the Pharmacopoeia for medical application [29]. However, the t-anethole content in FEO is dependent on its geographical origin and ranges from 38.8% (Albania, Yugoslavia) to 75.5% and 79.9% (France and Argentina, respectively), up to 84.1% (Turkey and Albania) [30]. The content of t-anethole in the FEO tested in the present study was close to French and Argentinean FEO with a concentration of 60%; other concentrations were 8% for pinene, 4% for estragole, and 6% for phencone, in accordance with the minimum concentrations required by the European Pharmacopoeia for medical use.

Many pharmacological effects of GEO, such as antibacterial, antifungal, analgesic, and anti-inflammatory, were already verified and confirmed both in experimental and preclinical studies, and contextually its safety is well-documented, and it is generally regarded as safe [31]. Although preliminary, our results demonstrate the efficacy of TEO, MEO, and LEO as valid alternatives to the use of antibiotics, whose frequent use can cause not only problems in the development of AMR but also a predisposition to secondary yeast infection.

EOs, already proposed for the treatment of human infections including cystitis, could also be evaluated in veterinary medicine and could represent an important natural alternative for the development not only of functional biomaterial, such as antimicrobial nanomaterials for catheters, but also of disinfectant and other medical devices [32]. Moreover, a study conducted by Mirskaya and Agranovski [33] demonstrated a quite broad range of efficiency of EOs in the control of biological aerosols, highlighting their potential use as new procedures and devices for the effective disinfection of surfaces and indoor air.

Although our results appear different from the literature, it is well known that the different quantitative composition of an EO and the synergistic effects among various components can contribute to a different antimicrobial activity. Furthermore, several studies indicate that the method of extraction, the season, and the geographical distribution can modify the composition of an EO and affect its antimicrobial properties [34]. At the same time, it is evident how difficult it is to define a reference standard to evaluate the effectiveness of an EO.

## 4. Materials and Methods

### 4.1. Essential Oil

The pure EOs of *Citrus Lemon* (LEO), *Pinus Sylvestris* (PEO), *Foeniculum Vulgare* (FEO), *Ocimum Basilicum* (BEO), *Melissa Officinalis* (MEO), *Thymus Vulgaris* (TEO), and *Zingiber Officinalis Rosc.* (GEO) were provided by Specchiasol S.r.l. (Bussolengo, VR, Italy) and were stored in a brown glass bottle at a temperature of 0–4 °C. Solvents (analytical grade), *n*-alkanes standard mixture C10-C40, and all standard compounds were purchased from Supelco Sigma-Aldrich S.r.l. (Milano, Italy). Filters were supplied by Agilent Technologies Italia S.p.a (Milano, Italy).

### 4.2. Gas Chromatography/Mass Spectrophotometry (GC/MS)

The gas chromatographic analyses of the EOs were performed on an Agilent 6890 N gas chromatograph equipped with a 5973 N mass spectrometer, provided with a HP-5 MS (5% phenylmethylpolysiloxane, 30 m, 0.25 mm i.d., 0.1 μm film thickness; J & W Scientific, Folsom) capillary column. The following temperature programmer was used: 5 min at 60 °C, then 4 °C/min to 220 °C, then 11 °C/min to 280 °C, hold for 15 min, for a total run of 65 min. The injector and detector temperatures were 280 °C; the carrier gas was He; the flow rate was 1 mL/min; the split ratio was 1:50; the acquisition range was 29–400 *m*/*z* in electron-impact (EI) mode; and the ionization voltage was 70 eV [10].

### 4.3. Compound Identification

For chemical characterization, the EOs were diluted 1:100 in ethyl acetate and after filtration, 1 μL of each EO solution was injected into the GC-MS. Qualitative analyses were carried out by comparing the calculated linear retention indices (LRIs) and similarity index of mass spectra (SI/MS) for the obtained peaks with the arithmetic index (AI) and the analogous data reported in the literature [18] and in the NIST 2017 databases (NIST 17, 2017. Mass Spectral Library—NIST/EPA/NIH. Gaithersburg, USA: National Institute of Standards and Technology. Last access 12_2021), respectively. The LRI of each compound was determined using temperature programming analysis and was calculated using the Van den Dool and Kratz equation [17] related to a homologous series of *n*-alkanes (C10–C40) under the same operating conditions. The SI/MS were determined as reported by Koo et al. [35]. 

The component relative percentages were calculated based on GC peak areas without using correction factors. Appendix A show a detailed description of EO chemotypes.

### 4.4. Bacteria Strains 

All the tests were performed on *E. coli* and *S. aureus*, ATCC strains, 25922 and 11622 (Manassas, VA, USA) respectively, and on two clinical isolates in the bacteriology laboratory of the Department of Veterinary Medicine, University of Bari, Italy. The two clinical samples were isolated from urine collected from two different dogs with recurrent cystitis characterized by morphological studies and identified by means of standard biochemical tests (API 20E and API 20 Staph System, BioMérieux, France) [36,37,38]. The biochemical analysis identified E. coli, protocol #462/20, and *S. aureus*, protocol #463/20. All the strains were stored at −20 °C until use in an appropriate culture medium, Tryptone Soya Broth (TSB) (Oxoid, Milan, Italy), with glycerol 20%.

Cultures for testing were prepared by inoculating 200 µL of each ATCC frozen organism and 2 colonies from each clinical isolate in 3 mL of Tryptic Soy Broth (TSB) and then incubating for 24 h at 37 °C. For the test, an inoculum concentration of 10^6^ CFU/mL from each culture was used according to the National Committee for Clinical Laboratory Standards, USA [39]. 

### 4.5. Screening for MDR Activity

Thirteen different antibiotics (Ampicillin-AMP, 10 μg; Amoxicillin + Clavulanic Acid-AMC,30 μg; Oxacillin-OX, 1 μg; Cephalexin-CL, 30 μg; Cefuroxime-CXM, 30 μg; Ceftriaxon-CRO, 30 μg; Cefotaxime-CTX, 30 μg; Doxycycline-DX, 30 μg; Gentamicin-GN, 30 μg; Erythromycin-E,15 μg; Co-Trimoxazole-SXT, 25 μg; Imipenem-IMI, 10 μg; Enrofloxacin-ENR, 15 μg) were used to investigate the in vitro antimicrobial activity of the two clinical samples, #462/20 and #463/20, using the disk diffusion method (DDM). The antibiotics were selected according to the standardized therapeutic protocols available for infection sustained by Gram-negative and Gram-positive bacteria according to Clinical & Laboratory Standards Institute (CLSI) guidelines. *E. coli* ATCC 25922 and *S. aureus* ATCC 11622 were used for quality control.

### 4.6. Screening for Antibacterial Activity of EOs 

The MIC and MBC were used to investigate the potential antibacterial activity of the selected EOs against the two different *E. coli* isolates (ATCC 25922 and clinical isolate #462/20) and two different *S. aureus* isolates (ATCC 11622 and clinical isolate #463/20) according to CLSI [40].

The MIC and MBC were tested with the method described by Moghimi et al. [41] modified as described. Broth microdilution assays were determined in 96-microtitration well plates (Greiner bio-one, Frickenhausen, Germany). Each EO was diluted in Mueller Hinton broth (MHb) with 2% dimethyl sulfoxide (DMSO) and phosphate buffer saline (PBS), pH 7.2, (ratio 1:8) to facilitate solubility in the culture medium, starting from the 40% concentration (*v*/*v*) down to 1.25% (*v*/*v*) in final volume (or, expressed in *w*/*v*, from 400 µg/mL down to 5 µg/mL). Each solution was tested in triplicate. Negative (MH broth with DMSO: PBS) and positive (MH broth with DMSO: PBS and bacterial inoculum, without EO) controls were prepared for each plate in columns 11 and 12, respectively. The plates were incubated at 37 °C for 24 h. The MIC value was determined as the lowest dilution where no bacterial growth was observed. The MBC was determined by subculturing 100 μL from each negative well of the plate into plate-count agar (PCA) plates. The MBC was defined as the lowest concentration in a subculture that tested negative or had the presence of only one colony after 24 h of incubation. 

To evaluate the potential effect of OEs on microbial growth, the CFU count was conducted from each well of plates showing bacterial growth. Briefly, 20 µL of supernatant were used to perform fourteen serial ten-fold dilutions in sterile saline solution (0.9% NaCl) and each dilution was included into plates containing PCA and incubated for 24 h at 37 °C. Dilutions that exhibited growth of more than 10 colonies were considered to evaluate the antibacterial efficacy of each EO [42].

### 4.7. Data Analysis

Statistical analysis was performed using the software R version 4.0.2 (R Foundation for Statistical Computing, Vienna, Austria; https://www.R-project.org/, access on 12 July 2021). A *p*-value < 0.05 was considered as statistically significant. The CFU measured for dilution of each EO were analyzed as continuous quantitative variables, and the normality distribution was evaluated using the Shapiro–Wilk normality test. Sperman’s correlation index (rho) was calculated to evaluate the correlation between different concentrations of the seven EOs tested and the CFU of each ATCC and clinical isolate strain of the two bacteria analyzed. The GC/MS analysis of each EO was replicated three times. The statistical analysis for chemical determination of structural equation modeling (SEM) was performed using Microsoft Excel.

## 5. Conclusions

The present study highlights how the bactericidal and bacteriostatic activity of the seven different EOs tested—LEO, PEO, FEO, BEO, MEO, TEO, and GEO—are related to the type of EO, the specific composition of each, and their concentration. Moreover, the species of bacterial strain tested influenced the activity of a single EO for possible different sensitivities within the same species.

Further studies on the efficacy, cytotoxicity, and structure analysis of these compounds with regard to bactericidal effect could be a valuable aid for pharmaceutical companies and researchers in the synthesis of natural antimicrobial drugs, ointments, and disinfectants to be used for sanitizing environments and the treatment of infections, both in humans and animals; this would represent an important natural alternative to the use of antibiotics.

## Figures and Tables

**Figure 1 antibiotics-11-00979-f001:**
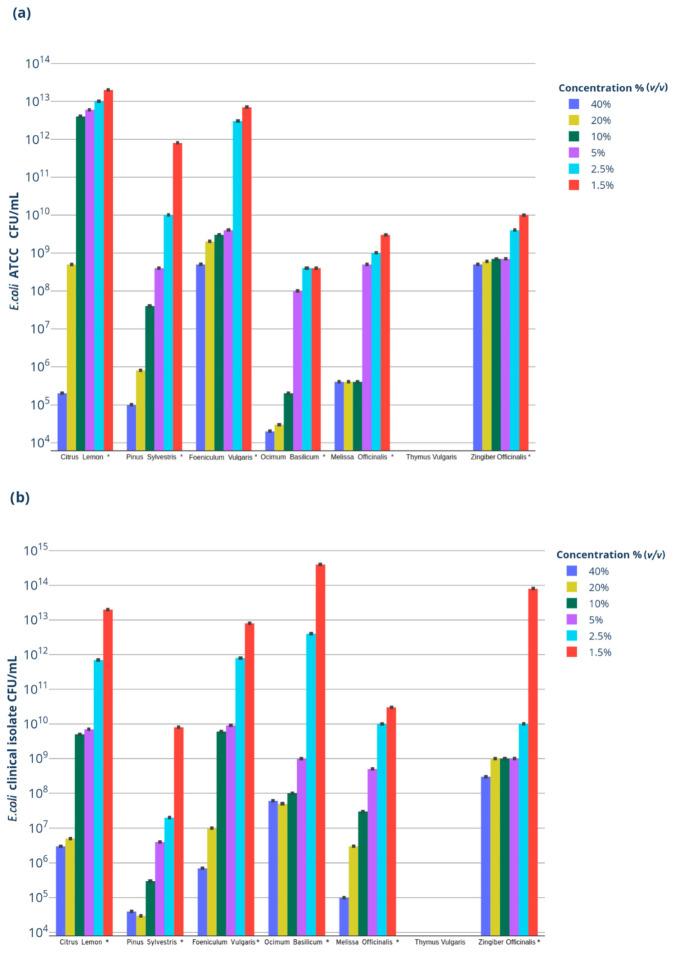
Bacterial growth of the tested strains. Response of bacterial strains in MH broth with varying percentages (*v*/*v*) of different EOs after 24 h of incubation, expressed as the rate of growth reduction in CFU/mL compared to positive controls. *: statistically significant dose–effect correlation (**a**) *E. coli* ATCC; (**b**) *E. coli* clinical isolate 462/20; (**c**) *S. aureus* ATCC; (**d**) *S. aureus* clinical isolate 463/20.

**Table 1 antibiotics-11-00979-t001:** Main components of tested Eos.

N	Components	AI	LEO	PEO	FEO	BEO	MEO	TEO	GEO
			% ± SEM	% ± SEM	% ± SEM	% ± SEM	% ± SEM	% ± SEM	% ± SEM
1	α-pinene ^a^	931	2.4 ± 0.5	29 ± 3	6.4 ± 0.9		0.34 ± 0.04	1.81 ± 0.10	1.73 ± 0.12
2	camphene ^a^	952		1.6 ± 0.3	0.12 ± 0.01		0.52 ± 0.05	1.89 ± 0.11	4.74 ± 0.27
3	β-thujene	968	1.94 ± 0.20	1.24 ± 0.23			0.20 ± 0.01	0.71 ± 0.06	
4	β-pinene ^a^	980	14.5 ± 1.0	17.2 ± 1.2	0.65 ± 0.07		0.65 ± 0.05	0.56 ± 0.03	
5	α-phellandrene ^a^	1003			5.2 ± 0.6			0.15 ± 0.01	4.4 ± 0.6
6	3-carene ^a^	1016		13.1 ± 2.5					
7	o-cymene	1021		1.1 ± 0.8	1 ± 0.1			19.6 ± 1.5	
8	eucalyptol ^a^	1023				0.29 ± 0.02	1.2 ± 0.5	0.89 ± 0.05	1.9 ± 0.4
9	limonene ^a^	1032	53 ± 5	9.8 ± 1.2	5.1 ± 1		4.3 ± 1	0.60 ± 0.04	
10	γ-terpinene ^a^	1064	5.9 ± 1.0		0.15 ± 0.02			9 ± 1	
11	β-linalool ^a^	1101	0.21 ± 0.02			17 ± 2.6	0.96 ± 0.25	4 ± 1	
12	endo-borneol ^a^	1167						1.8 ± 0.7	1.01 ± 0.12
13	estragole ^a^	1198			1.5 ± 0.1	73 ± 6			
14	citral ^a^	1240	3.8 ± 0.9			1.1 ± 0.08	43 ± 3		
15	geraniol	1254					2 ± 1		
16	anethole ^a^	1284			58.7 ± 3.9				
17	bornylacetate ^a^	1289		5.7 ± 1.3					
18	thymol ^a^	1290						47 ± 3	
19	geranyl aceate	1385	0.87 ± 0.06				1.95 ± 0.10		
20	caryophyllene ^a^	1415	0.136 ± 0.012	4.9 ± 0.9		0.43 ± 0.02	25 ± 1	2.2 ± 0.9	
21	α-bergamotene	1430	0.212 ± 0.020		0.105 ± 0.012	3.2 ± 0.4	0.14 ± 0.01		
22	humulene	1451		0.47 ± 0.02		0.237 ± 0.023	4.4 ± 0.9		
23	α-curcumene ^a^	1481							15 ± 1
24	zingiberene ^a^	1493							32.1 ± 1.8
25	β-sesquiphellandrene ^a^	1521							11 ± 1
26	caryophylleneoxyde	1592	0.31 ± 0.05	1.68 ± 0.29		0.223 ± 0.012	2.2 ± 0.9	0.58 ± 0.03	

^a^: Standard compounds. Arithmetic index (AI) was taken from Adams (2007) [17,18] and/or the NIST 2017 database. % ± SEM: relative percentage values of main compounds are means of three determinations with structural equation modeling (SEM) in all cases below 10%. Acronyms: LEO: *Citrus Lemon*; PEO: *Pinus Sylvestris*; FEO: *Feoniculum Vulgare*; BEO: *Ocimum Basilicum*; MEO: *Melissa Officinalis*; TEO: *Thymus Vulgaris*; GEO: *Zingiber Officinalis Rosc*.

**Table 2 antibiotics-11-00979-t002:** Bacterial growth reduction rate, expressed as log_10_ CFU/mL compared to positive control, based on different EO concentrations. (**a**) *E. coli* ATCC growth reduction rate; (**b**) *E. coli* clinical isolate 462/20 reduction rate; (**c**) *S. aureus* ATCC reduction rate; (**d**) *S. aureus* clinical isolate 463/20 reduction rate. Acronyms: EO: Essential Oil; LEO: *Citrus Lemon*; PEO: *Pinus Sylvestris*; FEO: *Feoniculum Vulgare*; BEO: *Ocimum Basilicum*; MEO: *Melissa Officinalis*; TEO: *Thymus Vulgaris*; GEO: *Zingiber Officinalis Rosc*.; *: CFU/mL; n.g.: no growth; n.i. = no inhibition.

(a)							
EO Concentration % (*v*/*v*)	LEO *	PEO *	FEO *	BEO *	MEO *	TEO *	GEO *
**40%**	10^11^	10^11^	10^7^	10^12^	10^11^	n.g.	10^8^
**20%**	10^8^	10^11^	10^7^	10^12^	10^11^	n.g.	10^8^
**10%**	10^4^	10^9^	10^7^	10^11^	10^11^	n.g.	10^8^
**5%**	10^4^	10^8^	10^7^	10^8^	10^8^	n.g.	10^8^
**2.50%**	10^3^	10^8^	10^4^	10^8^	10^9^	n.g.	10^7^
**1.25%**	10^1^	10^4^	10^4^	10^8^	10^9^	n.g.	10^6^
(**b**)							
**EO** **Concentration % (*v*/*v*)**	**LEO** *****	**PEO** *****	**FEO** *****	**BEO** *****	**MEO** *****	**TEO** *****	**GEO** *****
**40%**	10^8^	10^10^	10^9^	10^7^	10^9^	n.g.	10^6^
**20%**	10^8^	10^10^	10^7^	10^7^	10^7^	n.g.	10^5^
**10%**	10^5^	10^9^	10^5^	10^5^	10^7^	n.g.	10^5^
**5%**	10^5^	10^8^	10^5^	10^5^	10^7^	n.g.	10^5^
**2.50%**	10^3^	10^7^	10^5^	10^2^	10^4^	n.g.	10^4^
**1.25%**	10^1^	10^6^	10^4^	n.i.	10^3^	n.g.	10^1^
(**c**)							
**EO** **Concentration % (*v*/*v*)**	**LEO** *****	**PEO** *****	**FEO** *****	**BEO** *****	**MEO** *****	**TEO** *****	**GEO** *****
**40%**	n.g.	n.g.	n.i.	n.g.	n.g.	n.g.	10^11^
**20%**	n.g.	n.g.	n.i.	n.g.	n.g.	n.g.	10^11^
**10%**	n.g.	n.g.	n.i.	n.g.	n.g.	n.g.	10^11^
**5%**	n.g.	n.g.	n.i.	n.g.	n.g.	n.g.	10^13^
**2.50%**	n.g.	n.g.	n.i.	n.g.	n.g.	n.g.	10^14^
**1.25%**	n.g.	10^2^	n.i.	n.g.	n.g.	n.g.	10^14^
(**d**)							
**EO** **Concentration % (*v*/*v*)**	**LEO** *****	**PEO** *****	**FEO** *****	**BEO** *****	**MEO** *****	**TEO** *****	**GEO** *****
**40%**	n.g.	n.g.	n.i.	n.g.	n.g.	n.g.	10^12^
**20%**	n.g.	n.g.	n.i.	n.g.	n.g.	n.g.	10^12^
**10%**	n.g.	n.g.	n.i.	n.g.	n.g.	n.g.	10^12^
**5%**	n.g.	n.g.	n.i.	n.g.	n.g.	n.g.	10^13^
**2.50%**	10^12^	n.g.	n.i.	n.g.	n.g.	n.g.	10^14^
**1.25%**	10^13^	10^2^	n.i.	n.g.	n.g.	n.g.	10^14^

## Data Availability

Not applicable.

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
