# Peer review of "Antimicrobial Activity of Essential Oils Evaluated In Vitro against Escherichia coli and Staphylococcus aureus"

_antibiotics, 2022, doi:10.3390/antibiotics11070979_

Round 1
Reviewer 1 Report
The manuscript describes the chemical composition of different commercial essential oils and their evaluation against four pathogens: two strains of S. aureus and two strains of E. coli. Although the study is preliminary, the results could be supportive regarding the bactericidal and bacteriostatic activity. From a scientific point of view, I have a few comments and suggestions:
Can the authors provide a chromatogram for chemical composition of each condition? What is the percentage of each component?
Lines 140-142; please explain "E. coli isolate #462/20". A reference should be given for this protocol. Also for #463/20
Please explain the superscripts in Table 1. This Table (or Figure 1) can be included as Sup Info. The quality of Figure 1 could be improved. Moreover, neither standard deviation nor statistical significance is provided.
Can the authors discuss about the potential mechanism of these compounds (cell wall synthesis/membrane, protein synthesis inhibitors?)
Author Response
Dear Reviewer,
thank you for giving us the opportunity to submit a revised draft of our manuscript titled: “Antimicrobial activity of essential oils evaluated in vitro against Escherichia coli and Staphylococcus aureus”, Manuscript ID: antibiotics-1806348, to Antibiotics.
We appreciate the time and effort that you have dedicated to providing your valuable feedback on our manuscript. We are grateful to the reviewers for their insightful comments on our paper. We have been able to incorporate changes to reflect the suggestions provided by the editor and reviewers. We have highlighted the changes within the manuscript.
Here is a point-by-point response to your comments and concerns.
Comments from Reviewer 1 (R1)
The manuscript describes the chemical composition of different commercial essential oils and their evaluation against four pathogens: two strains of S. aureus and two strains of E. coli. Although the study is preliminary, the results could be supportive regarding the bactericidal and bacteriostatic activity. From a scientific point of view, I have a few comments and suggestions:
R1.1: Can the authors provide a chromatogram for chemical composition of each condition? What is the percentage of each component?
Reply to R1.1: Essential oils chemotype and composition was performed as reported in the section “Materials and Methods” using GC/MS techniques. The chromatograms and printouts of all the essential oils used in this work are available for the Reviewers and Editors but are strictly confidential. We are not authorized by the manufacturers to distribute them in that format. In table 1 and in tables S1 of supporting files the chemical composition and relative percentage of each EOs component are reported in the columns titled %±SEM.
R1.2: Lines 140-142; please explain "E. coli isolate #462/20". A reference should be given for this protocol. Also for #463/20
Reply to R1.2: As reported in the text, E. coli isolate #462/20 and S. aureus isolate #463/20 are two strains isolated from urinary clinical samples of two dogs, with recurrent cystitis, in the laboratory of the Department of Veterinary Medicine, University of Bari, Italy. After the isolation, these strains were characterized by morphological studies and identified using standard biochemical tests (API 20E and API 20 Staph System, BioMérieux, France). No references are available for these two strains because they are strains isolated during routinely diagnostic activities and represent our positive controls in the field.
R1.3: Please explain the superscripts in Table 1. This Table (or Figure 1) can be included as Sup Info. The quality of Figure 1 could be improved. Moreover, neither standard deviation nor statistical significance is provided.
Reply to R1.3: The superscript has been modified as “Table 1. Main components of tested EOs” as suggested. Table 1 lists only the main components of each essential oil tested and could be useful for the readers to quickly recognizing the chemotype of each essential oil and all the components that are common among all essential oils. While, in tables S1 of supporting files all the components identified in each essential oil are reported.
According to the reviewer’s suggestion, Figure 1 was edited, including the standard deviation and statistical significance information.
R1.4: Can the authors discuss about the potential mechanism of these compounds (cell wall synthesis/membrane, protein synthesis inhibitors?)
Reply to R1.4: The potential mechanisms of chemical compounds of EOs were briefly discussed in the text in lines 260-263 just for GEO which showed a dose-dependent antimicrobial activity against all bacterial strains tested in this study. However, the study aims to evaluate the potential antibacterial activity of seven EOs at different concentrations against two strains of S. aureus (ATCC 11622 and urinary bacterial isolate) and two strains of E. coli (ATCC 25922 and urinary bacterial isolate). Since the antimicrobial activity of many essential oils is due to the synergistic action of their components, we did not consider the activity of the individual chemical components or their mechanism of action on the individual bacterial strain, but the overall activity of the essential oil. Given the extreme variability of the composition of essential oils, influenced by many factors, the chemical composition of the single EOs was evaluated to compare our data with the technical data sheets of the manufacturing companies and therefore to have information regarding the purity of the individual active ingredients.
Reviewer 2 Report
Manuscript titled “Antimicrobial activity of essential oils evaluated in vitro against Escherichia coli and Staphylococcus aureus” reports the use of some essential oils, and their effectiveness against E. coli and S. aureus. The work is relevant, according to the ubiquity of the organisms studied in human health and disease, and the various natural oils used as potential antimicrobials. The introduction is brief and concise, results and discussion are mostly adequate, but there are some comments for the authors to improve their document:
1. In the abstract, please mention some results that indicate the most relevant knowledge reported in the manuscript. For example, what compounds were the most abundant, which oil was the most effective etc. Some numerical data may serve to better illustrate this.
2. Please homogenize the number of decimal digits for all values listed in table 1, for example, the value of α-pinene is 2.42±0.520 (LEO column), both the value and its error should have the same precision.
3. In table 2, it is unnecessary to repeat “CFU/mL” for all values. Please consider stating these units once atop the table or on its legend, in order to make it easier to read.
4. In lines 229-235, some values should be in superscript (i.e. “101” instead of “101”).
5. Authors mention that their results are similar to those reported in reference 21 (line 254). Please briefly mention what results are reported there that are similar to those reported herein, in order to make the comparison clearer.
6. In line 286, authors mention that their results conflict with the literature, but it can be explained due to “their functional groups in the active components”. Does this refer to isomerism or substitution of the main molecules? Or perhaps the authors mean that the particular composition of the oil (say 10 % of compound A and 90 % of compound B) is the one that varies by source. If the latter interpretation is correct, it was already stated in the previous line, where oil composition is specifically mentioned. If the authors are indeed suggesting molecular variation of some compounds, please provide some specific examples to better substantiate this argument.
7. In the conclusion, please consider making it more concise and avoid additional citations here. Some arguments are better suited for the discussion instead. The conclusion should clearly state what was actually done in the present work and the main knowledge reported (similar to comment 1 for the abstract).
Author Response
Dear Reviewer,
thank you for giving us the opportunity to submit a revised draft of our manuscript titled: “Antimicrobial activity of essential oils evaluated in vitro against Escherichia coli and Staphylococcus aureus”, Manuscript ID: antibiotics-1806348, to Antibiotics.
We appreciate the time and effort that you have dedicated to providing your valuable feedback on our manuscript. We are grateful to the reviewers for their insightful comments on our paper. We have been able to incorporate changes to reflect the suggestions provided by the editor and reviewers. We have highlighted the changes within the manuscript.
Here is a point-by-point response to your comments and concerns.
Comments from Reviewer 2 (R2)
Manuscript titled “Antimicrobial activity of essential oils evaluated in vitro against Escherichia coli and Staphylococcus aureus” reports the use of some essential oils, and their effectiveness against E. coli and S. aureus. The work is relevant, according to the ubiquity of the organisms studied in human health and disease, and the various natural oils used as potential antimicrobials. The introduction is brief and concise, results and discussion are mostly adequate, but there are some comments for the authors to improve their document:
R2.1: In the abstract, please mention some results that indicate the most relevant knowledge reported in the manuscript. For example, what compounds were the most abundant, which oil was the most effective etc. Some numerical data may serve to better illustrate this.
Reply to R2.1: We agree with the reviewer’s comment and the abstract was edited. Indeed, we added more information on the results regarding the major components of all seven tested oils tested, and their bactericidal/bacteriostatic activity related to different concentrations.
R2.2: Please homogenize the number of decimal digits for all values listed in table 1, for example, the value of α-pinene is 2.42±0.520 (LEO column), both the value and its error should have the same precision.
Reply to R2.2: We agree with the reviewer’s comment. The number of decimal digits for all values listed in table 1 and in tables S1 of supporting files were homogenized to the same precision.
R2.3: In table 2, it is unnecessary to repeat “CFU/mL” for all values. Please consider stating these units once atop the table or on its legend, in order to make it easier to read.
Reply to R2.3: We agree with the reviewer’s comment. Following, the suggestions of the referee we edited the Table 2 in revised manuscript.
R2.4: In lines 229-235, some values should be in superscript (i.e. “101” instead of “101”).
Reply to R2.4: After carefully revising the text, we edited the manuscript according to the reviewer's comment.
R2.5: Authors mention that their results are similar to those reported in reference 21 (line 254). Please briefly mention what results are reported there that are similar to those reported herein, in order to make the comparison clearer.
Reply to R2.5: The reviewer's observation is pertinent, and we thank him for pointing out the error. Reference 21 is not correct. In the modified text, we have inserted the proper references by adding what is requested in the comment see lines 246-253.
R2.6: In line 286, authors mention that their results conflict with the literature, but it can be explained due to “their functional groups in the active components”. Does this refer to isomerism or substitution of the main molecules? Or perhaps the authors mean that the particular composition of the oil (say 10 % of compound A and 90 % of compound B) is the one that varies by source. If the latter interpretation is correct, it was already stated in the previous line, where oil composition is specifically mentioned. If the authors are indeed suggesting molecular variation of some compounds, please provide some specific examples to better substantiate this argument.
Reply to R2.6: We agree with the observation of Reviewer 2. Following the referee's suggestion, we have modified the revised manuscript, rephrasing the sentence to clarify concepts expressed in the text. The authors mean that the particular composition of the oils varies by source, geographical distribution and season (lines 299-305)
R2.7: In the conclusion, please consider making it more concise and avoid additional citations here. Some arguments are better suited for the discussion instead. The conclusion should clearly state what was actually done in the present work and the main knowledge reported (similar to comment 1 for the abstract).
Reply to R2.7: We agree with the reviewer's comment. Following the referee's suggestions, we edited the "Conclusions" section in the revised manuscript. We have sanities the conclusions and moved some previously reported information in this section to the "Discussion."
Round 2
Reviewer 2 Report
Manuscript titled “Antimicrobial activity of essential oils evaluated in vitro against Escherichia coli and Staphylococcus aureus” reports the use of some essential oils, and their effectiveness against E. coli and S. aureus. The study reports interesting information about the effects of various oils against common microorganisms involved in many pathologies, thus, it can be considered of relevance to the scientific literature.
The most recently submitted version of the manuscript was revised according to comments and suggestions made during an initial revision. Among those made by the present reviewer include:
1. Mentioning some results that illustrate the most relevant knowledge reported in the manuscript. The abstract was modified accordingly, where some results are now clearly mentioned, in particular, regarding oil composition and a dose-effect correlation against microorganisms studied.
2. Homogenizing the number of decimal digits for the data shown in table 1. Some corrections were made, but there are still some values where the precision of a value and its error differs. Please carefully revise those values to the same precision.
3. Avoid repetition of “CFU/mL” for all values listed in Table 2. The correction was made to avoid unnecessary repetition and increase the clarity of this table.
4. Fixing the superscript for some values in the main text. The change was made accordingly.
5. Providing more information about the results reported by other authors which are similar to those of the present work. The phrasing of the sentence in question was modified to clarify the authors’ intended meaning, and a mistake on the reference was amended.
6. Clarifying the authors’ intention when they suggest that their results conflict with the literature. The sentences have been edited to reflect how the composition of an oil will differ according to various factors, and how these changes will be reflected on their ability as antibacterial agents.
7. Making the conclusion more concise, avoid additional citations and moving some arguments to the discussion. The conclusion is now shorter and precisely indicates what knowledge has been gained from the present work; citations have also been removed from this section and some text moved to the discussion.
According to the aforementioned changes made to the document, it is apparent that most recommendations made were properly addressed by the authors. The present reviewer has no additional comments or suggestions to warrant an additional review, but instead recommends that the precision of the data is homogenized in table 1 before publication, as mentioned in comment 2.
Author Response
Reviewer comment: The present reviewer has no additional comments or suggestions to warrant an additional review, but instead recommends that the precision of the data is homogenized in table 1 before publication, as mentioned in comment 2 (Homogenizing the number of decimal digits for the data shown in table 1. Some corrections were made, but there are still some values where the precision of a value and its error differs. Please carefully revise those values to the same
Reply: The number of decimal digits for all values listed in table 1 and in the text were homogenized to the same precision according with literature reference: J. F. Caballero, D.F. Harris; There seems to be Uncertainty about the use of significant figures in reporting Uncertainties of results. Journal of Chemical Education, 996, vol 75 No.8 (1998)
Please see the attachment.
